# The Q163C/Q309C mutant of $\alpha_M$I-domain is an active variant suitable for NMR characterization

Hoa Nguyen[1], Tianwei Jing[2], Xu Wang[1]*

1 School of Molecular Sciences, Arizona State University, Tempe, Arizona, United States of America,
2 Biosensing Instrument Inc., Tempe, Arizona, United States of America

* xuwang@asu.edu

## Abstract

Integrin $\alpha_M\beta_2$ (Mac-1, CD11b/CD18, CR3) is an important adhesion receptor expressed on monocytes. Mac-1 is responsible for mediating cell migration, phagocytosis, degranulation as well as cell-cell fusion. It is also the most promiscuous integrin in terms of ligand specificity with over 100 ligands, most of which use the $\alpha_M$I-domain as their binding site. Despite the importance of $\alpha_M$I-domain in defining ligand interactions of Mac-1, structural studies of $\alpha_M$I-domain's interactions with ligands are lacking. In particular, solution NMR studies of $\alpha_M$I-domain's interaction with ligands have not been possible because the most commonly used active $\alpha_M$I-domain mutants (I316G and ΔK315) are not sufficiently stable and soluble to be used in solution NMR. The goal of this study is to identify an $\alpha_M$I-domain active mutant that's amenable to NMR characterization. By screening known activating mutations of $\alpha_M$I-domain, we determined that the Q163C/Q309C mutant, which converts the $\alpha_M$I-domain into its active form through the formation of an intramolecular disulfide bond, can be produced with a high yield and is more stable than other active mutants. In addition, the Q163C/Q309C mutant has better NMR spectral quality than other active mutants and its affinity for ligands is comparable to other active mutants. Analysis of the $Co^{2+}$-induced pseudocontact shifts in the Q163C/Q309C mutant showed the structure of the mutant is consistent with the active conformation. Finally, we show that the minor fraction of the Q163C/Q309C mutant without the disulfide bond can be removed through the use of carboxymethyl sepharose chromatography. We think the availability of this mutant for NMR study will significantly enhance structural characterizations of $\alpha_M$I-domain-ligand interactions.

## Introduction

Integrins are heterodimeric adhesion receptors ubiquitous to all metazoans [1]. Integrins play vital roles in numerous cellular processes including cell growth, proliferation, differentiation, and migration. In the cells of the innate immune system, integrins are also responsible for the phagocytosis of opsonized targets as well as fusion with other immune cells [2, 3]. Much of integrins' activities depend on their ability to bind specifically to extracellular proteins. As a

**Data Availability Statement:** Chemical shift assignments of the Mg2+ species of the Q163C/Q309C mutant have been deposited in the BMRB with the BMRB ID 51714. Chemical shifts for the Co2+ species of the Q163C/Q309C mutant have

been deposited in the BMRB with the BMRB ID 51716.

**Funding:** The study was funded by a grant from the National Institutes of Health to XW (www.nih. gov, grant #R01GM118518). NIH did not play any role in the study design, data collection and analysis, decision to publish, or preparation of the manuscript.

**Competing interests:** The authors have declared that no competing interests exist.

result, ligand specificities of integrins are vital determinants of their activities, and understanding the ligand binding mechanisms behind these specificities is an essential step in the development of therapies targeting integrins.

All integrins contain an α and a β subunit. Humans have 18 different α subunits and 8 different β subunits. The selective pairing of these subunits produces 24 unique integrin receptors in humans. Different integrins recognize different ligands [4]. The ligand specificity of the myeloid cell integrin $\alpha_M\beta_2$ (Mac-1, CD11b/CD18, CR3) is especially interesting. Mac-1 is expressed primarily on myeloid cells such as neutrophils and macrophages. It is involved in the phagocytosis of opsonized particles, cell-cell fusion, and acts as an important regulator of the immune activities of these cells [5]. Its ligand specificity is also the broadest among all integrins. So far, more than 100 ligands have been reported to bind to Mac-1 [2]. Mac-1 ligands vary greatly in chemical structure and physical properties. However, the binding site for the vast majority of ligands is located in the I-domain of Mac-1's $\alpha_M$ subunit ($\alpha_M$I-domain). Despite the importance of $\alpha_M$I-domain in ligand binding, only the structure of $\alpha_M$I-domain with one of its natural ligands, the complement fragment iC3b, is determined [6, 7]. The mechanism by which $\alpha_M$I-domain recognizes most other ligands remains unknown [2, 3]. An important factor contributing to the lack of understanding of $\alpha_M$I-domain's ligand specificity is the absence of an appropriate active $\alpha_M$I-domain variant suitable for solution NMR characterization, a major tool for structural biologists.

$\alpha_M$I-domain is a ~ 200 residue domain with a Rossmann fold. The apical side of the domain contains a metal-binding site capable of chelating divalent cations such as $Mg^{2+}$, $Ca^{2+}$, $Co^{2+}$, and $Mn^{2+}$. These metals are usually crucial mediators of ligand binding. As a result, the metal-binding site is referred to as the metal ion-dependent adhesion site (MIDAS). Similar to other integrins, Mac-1 can adopt both inactive and active conformations. In the active conformation, the C-terminal α7 helix of $\alpha_M$I-domain is pulled away from the MIDAS, causing the metal ion coordination to change. This allows the metal ion to be coordinated by acidic amino acids on the ligand, thereby greatly increasing the affinity of the domain for its ligands [8]. Because of active $\alpha_M$I-domain's higher affinity for ligands, most structural biology studies of $\alpha_M$I-domain's interactions with ligands use active $\alpha_M$I-domain. These active forms are generated from wild type, inactive $\alpha_M$I-domain by pulling the C-terminal helix away from the MIDAS using several methods. The most popular method is to either mutate residue I316 to Gly (I316G) or to truncate the C-terminal α7 helix at residue K315 (ΔK315) [8, 9]. This method disrupts the hydrophobic interactions that hold the α7 helix in place, allowing the domain to adopt the active conformation. Although these mutants have been successfully used in X-ray crystallography studies of $\alpha_M$I-domain's interactions with ligands [6–8, 10], there is no report on the suitability of these active mutants for use in solution NMR studies.

In this study, we show that both the I316G mutation and truncation at K315 lead to significant destabilization of $\alpha_M$I-domain, resulting in lower protein yield, lower solubility and lower NMR signal intensities. These disadvantages made solution NMR characterizations of $\alpha_M$I-domain's ligand interactions difficult. To explore the suitability of other active $\alpha_M$I-domain mutants for NMR studies, we examined three other active mutants. Unlike the I316G and ΔK315 mutants, these mutants force the $\alpha_M$I-domain into the active conformation by constraining the position of the α7 helix using disulfide bonds. In particular, McCleverty and Liddington reported that the addition of the intramolecular disulfide bond in the mutant D132C/ K315C enhanced its affinity for the ligand ICAM1 [11]. Shimaoka et al. also reported that the mutations Q163C/Q309C, and D294C/Q311C in $\alpha_M$ increased Mac-1-expressing HEK293T cells' affinity for iC3b in HEK293T cells [12] (see Fig 1 of S1 Fig for the positions of the cysteine mutations). However, no structural studies were done to confirm these $\alpha_M$I-domain mutants were in the active form. We hypothesize that, by not disrupting the hydrophobic core,

these active mutants should be more stable and more amenable to structural characterization by solution NMR. To confirm this hypothesis, we expressed and purified all three mutants and characterized them using NMR, SPR, and other biophysical techniques. Our results show that only one of the mutants (Q163C/Q309C) can spontaneously form the required disulfide bond. Although treatment of the D132C/K315C mutant with the oxidizing agent $Cu^{2+}$/phenanthroline was reported to produce active forms of the mutant, our data indicate only ~ 50% of protein were disulfide bonded after the treatment while unintended modifications associated with oxidations of other amino acids also took place. In addition, expressing the Q163C/Q309C mutant in the *E. coli* strain OrigamiB(DE3), which contains lower levels of the reductants glutathione and thioredoxin, further improved the percentage of proteins with the internal disulfide bond from ~ 90% to ~ 95%. More importantly, the Q163C/Q309C mutant has better yield, higher thermal stability, and better NMR spectral quality. Analysis of $Co^{2+}$-induced pseudocontact shifts (PCS) of the Q163C/Q309C mutant showed the structure of the mutant is consistent with the active conformation. SPR analysis also showed that the Q163C/Q309C mutant has a similar affinity for the ligand C3d as the I316G and ΔK315 mutants. Finally, the small fraction of protein without the intramolecular disulfide bond can be easily removed by taking advantage of the high affinity of the active $\alpha_M$I-domain for carboxymethyl dextran. In sum, these results show the Q163C/Q309C mutant is excellent for NMR studies. We think this mutant will be useful to researchers interested in studying the interactions of $\alpha_M$I-domain with its ligands.

## Materials and methods

### Expression and purification of $\alpha_M$I-domain

The expression and purification of $\alpha_M$I-domain followed the procedures in Feng et al. [13]. Briefly, the open reading frame (ORF) of the wild type human $\alpha_M$I-domain (E131-T324) was cloned into the pHUE vector [14] using *Sac*II and *Hind*III as restriction sites. Cysteine mutations (Q163C/Q309C, D132C/K315C, D294C/Q311C) were introduced into $\alpha_M$I-domain using the Q5 Site-Directed Mutagenesis Kit (NEB). The plasmids were transformed into either BL21(DE3) or OrigamiB(DE3) (Millipore Sigma) and the cells were grown in M9 media at 37˚C until the culture reached an $OD_{600}$ of ~ 0.8–1. The protein expression was induced with 0.5 mM IPTG, and cells were harvested after overnight incubation at 22˚C. Cells were lysed with sonication after a 20-minute incubation on ice in a buffer containing 20 mM sodium phosphate, pH 7, 0.5 M NaCl, 10 mM imidazole, 5% glycerol, and 1 mg/ml lysozyme. After centrifugation to remove insoluble material, the supernatant was passed through a 5-mL HisTrap column (Cytiva) and the protein was eluted with a 0.01 to 0.5 M gradient of imidazole. To separate the N-terminal His-Ubiquitin from $\alpha_M$I-domain, the protein was digested with enzyme USP2 (1:50 molar ratios) overnight at room temperature in 20 mM Tris, pH 8, 100 mM NaCl. The digestion mixture was then subjected to $Ni^{2+}$ column purification again. $\alpha_M$I-domain in the flow through was further purified using a 120-mL Superdex 75 column equilibrated in buffer containing 20 mM HEPES, 0.3 M NaCl, pH 7.0. Finally, the protein was exchanged to 20 mM HEPES, 100 mM NaCl, pH 7.0 for thermal shift and NMR analyses. SPR analysis was carried out in 20 mM HEPES, pH 7 buffer containing 0.1 M NaCl, 1 mM $MgCl_2$, and 0.05% Tween 20. $^{15}N$, $^{13}C$, and $^{2}H$ isotope enrichment was accomplished using $^{15}N$-enriched $NH_4Cl$, $^{2}H$,$^{13}C$-enriched glucose, and $D_2O$. In particular, OrigamiB(DE3) cells freshly transformed with the expression plasmid for the Q163C/Q309C mutant was first grown in LB at 37˚C until an $OD_{600}$ of ~1.0. 2 mL of the culture was then pelleted gently at room temperature and used to seed 50 mL of $^{2}H$,$^{13}C$,$^{15}N$-enriched minimal media containing 4 g/L of $^{2}H$,$^{13}C$-glucose. The culture was grown overnight at 37˚C and diluted with 450 mL of

fresh $^2$H,$^{13}$C,$^{15}$N-enriched minimal media. The large scale culture was grown to an OD$_{600}$ of ~ 0.8 to 1.0 and induced with 0.5 mM of IPTG. The culture was then placed in an shaker incubator at 22˚C for 18 hours before being harvested.

## Expression and purification of C3d

C3d was expressed and purified according to Bajic et al. [6]. Briefly, ORF of C3d (residues 993 to 1288) was cloned into pET15b with 6XHis and TEV cleavage site at the N-terminus and expressed in *E. coli* BL21(DE3) using a similar procedure as $α_M$I-domain. Specifically, after harvesting, cells were resuspended in 20 mM Tris, 200 mM NaCl, pH 8.0 in addition to 1 mg/ mL lysozyme. The supernatant after sonication was passed through a 5-ml HisTrap column (Cytiva) and the protein was eluted using a 0.01 to 0.5 M imidazole gradient. The chimera protein was digested with TEV protease with a protein-to-enzyme ratio of 50. The digested protein was subjected to a second Ni$^{2+}$-affinity column. Flow through fractions containing C3d were combined and concentrated.

## Protein thermal shift assay of $α_M$I-domain

To measure the thermal stability of $α_M$I-domain, 10 μg of $α_M$I-domain in 12.5 μL of 20 mM HEPES, 0.1 M NaCl, pH 7 buffer were mixed with 2.5 μL of 8X protein thermal shift dye (Thermo Fisher Scientific), and 5 μL of 4X protein thermal shift assay buffer (Thermo Fisher Scientific). Duplicates of each sample were heated from 22˚C to 99˚C with a temperature gradient of 0.015˚C / second in a QuantStudio 3 qPCR instrument. The fluorescence of the samples was measured using an excitation wavelength of 580 nm and an emission wavelength of 623 nm.

## SPR analysis of $α_M$I-domain's interaction with C3d

SPR analysis was carried out on a BI-4500 SPR instrument (Biosensing Instrument). To carry out the measurement, 50 μM C3d was flowed at a rate of 20 μL/min over EDC/NHS-activated CM-dextran sensor until a response of ~ 1000 RU was observed. An increase of 1 RU corresponds to protein deposition of ~ 1 pg / mm$^2$. After washing with 1.5 M NaCl, wild type, ΔK315, I316G, and Q163C/Q309C $α_M$I-domain at concentrations of 0.19, 0.39, 0.78, 1.56, 3.13, 6.25, and 12.5 μM were flowed over the sensor while data were acquired. The sensor was regenerated with 1.5 M NaCl after each sample. Response curves were background corrected by subtracting the response of the reference channel with no immobilized C3d.

## NMR data collection and analysis

NMR data were acquired on a Bruker 600 MHz instrument equipped with Avance III HD console and a Prodigy probe. All NMR samples contained 0.1 to 0.3 mM $^{15}$N-labeled $α_M$I-domain in 20 mM HEPES, 0.1 M NaCl, pH 7.0 buffer. To study the effect of Mg$^{2+}$ and glutamate on the protein, 1 mM MgCl$_2$ and 10 mM sodium glutamate were also included in the sample. $^{15}$N-edited HSQC spectra were acquired using the Bruker pulse sequence fhsqcf3gpph. Pseudocontact shifts (PCS) were measured by comparing the amide hydrogen chemical shifts of the $^2$H/$^{13}$C/$^{15}$N-labeled Q163C/Q309C mutant in the Co$^{2+}$-bound form with the Mg$^{2+}$-bound form. PCS samples also contained 10 mM glutamate to prevent aggregation of the protein. Backbone amide hydrogen and nitrogen assignments were obtained using information from HNCACB and HNCOCAB spectra for both Co$^{2+}$- and Mg$^{2+}$-bound proteins. To assign the chemical shifts of the Mg$^{2+}$ species, we first tabulated the HN, N, CA, CB chemical shifts as well as CA and CB chemical shifts of the previous amino acids for each spin system using the

spectral data. The information was then used to derive possible assignment using iPINE [15]. Each assignment was then examined manually to confirm the assignment. Similarly, we tabulated the HN, N, CA, CB chemical shifts as well as CA and CB of the previous amino acids for each spin system in the $Co^{2+}$ data. Then, utilizing the fact that PCS of backbone amide H and N are similar if the atoms are not too close to the paramagnetic center, possible assignments were proposed by finding assigned signals that are shifted diagonally in the HSQC spectrum of $Mg^{2+}$ sample. The possible assignments were further validated by comparing the CA and CB chemical shifts of the $Co^{2+}$ and $Mg^{2+}$ samples, taking into consideration that CA and CB chemical shifts may have PCS of similar magnitudes. The preliminary PCS tensor calculated using these assignments were used to predict additional assignments. NMR data were processed using NMRPipe [16] and analyzed using NMRViewJ [17]. Fitting of PCS to the structures of active and inactive $\alpha_M$I-domain (PDB ID 1IDO and 1JLM) was carried out using the

software Paramagpy [18]. The quality factor of fitting is defined as $\sqrt{\frac{\sum_i (a_i^{exp} - a_i^{cal})^2}{\sum_i (a_i^{exp})^2}}$, where $a_i^{exp}$

and $a_i^{cal}$ are the experimental and calculated PCS for atom i. Chemical shift assignments of the $Mg^{2+}$ species of the Q163C/Q309C mutant have been deposited in the BMRB with the BMRB ID 51714. Chemical shifts for the $Co^{2+}$ species of the Q163C/Q309C mutant have been deposited in the BMRB with the BMRB ID 51716.

## Results

### Spontaneous disulfide bond formation in mutants

Our study is motivated by a desire to study the ligand specificity of $\alpha_M$I-domain using solution NMR. In our hands, both the ΔK315 mutant (residues E131 to K315) and the I316G mutant showed low solubility and poor NMR spectral quality (*vide infra*). We attribute this to the fact that the disruption of the hydrophobic core of $\alpha_M$I-domain may have significantly destabilized the protein, rendering them unsuitable for solution NMR studies. Fortunately, activation of $\alpha_M$I-domain using constraining disulfide bonds that pull the C-terminal α7 helix away from MIDAS has also been reported. We recombinantly expressed three such mutants. These mutants are D132C/K315C [11], Q163C/Q309C [12], and D294C/Q311C [12]. Of these three mutants, D132C/K315C has been shown to have a higher affinity for ICAM1 after treatment with an oxidizing agent whereas HEK293T cells expressing the Q163C/Q309C, and D294C/Q311C Mac-1 mutants achieved higher ligand affinity even without oxidizing treatments [12]. We recombinantly expressed all three mutants in *E. coli* BL21(DE3). Analysis of the purified proteins using SDS-PAGE under reducing and non-reducing conditions revealed that the formation of the intramolecular disulfide bond significantly increased the migration distance of the protein in SDS-PAGE. This provided us with a simple method to determine the fraction of proteins containing the intramolecular disulfide bond. Fig 1A shows SDS-PAGE analysis of the three mutants in the presence and absence of 1 mM DTT. It is clear that neither the D132C/K315C nor the D294C/D311C mutant was able to form a significant amount of the intramolecular disulfide bond while close to 90% of the Q163C/Q309C mutant did. In addition, both the D132C/K315C and the D294C/D311C mutants produced a detectable amount of dimer as a result of intermolecular disulfide bonding, but no dimer was observed in the Q163C/Q309C mutant.

Because it was reported that the intramolecular disulfide bond in the D132C/K315C mutant only forms after the protein was treated with 0.1 μM $Cu^{2+}$/phenanthroline complex [11], we examined the efficiency of such a process. Besides 0.1 μM of $Cu^{2+}$/phenanthroline, we also tested the effect of 1 mM $MgCl_2$ and 10 mM sodium glutamate on the efficiency of disulfide bond formation since these compounds are ligands of active $\alpha_M$I-domain [19] and may

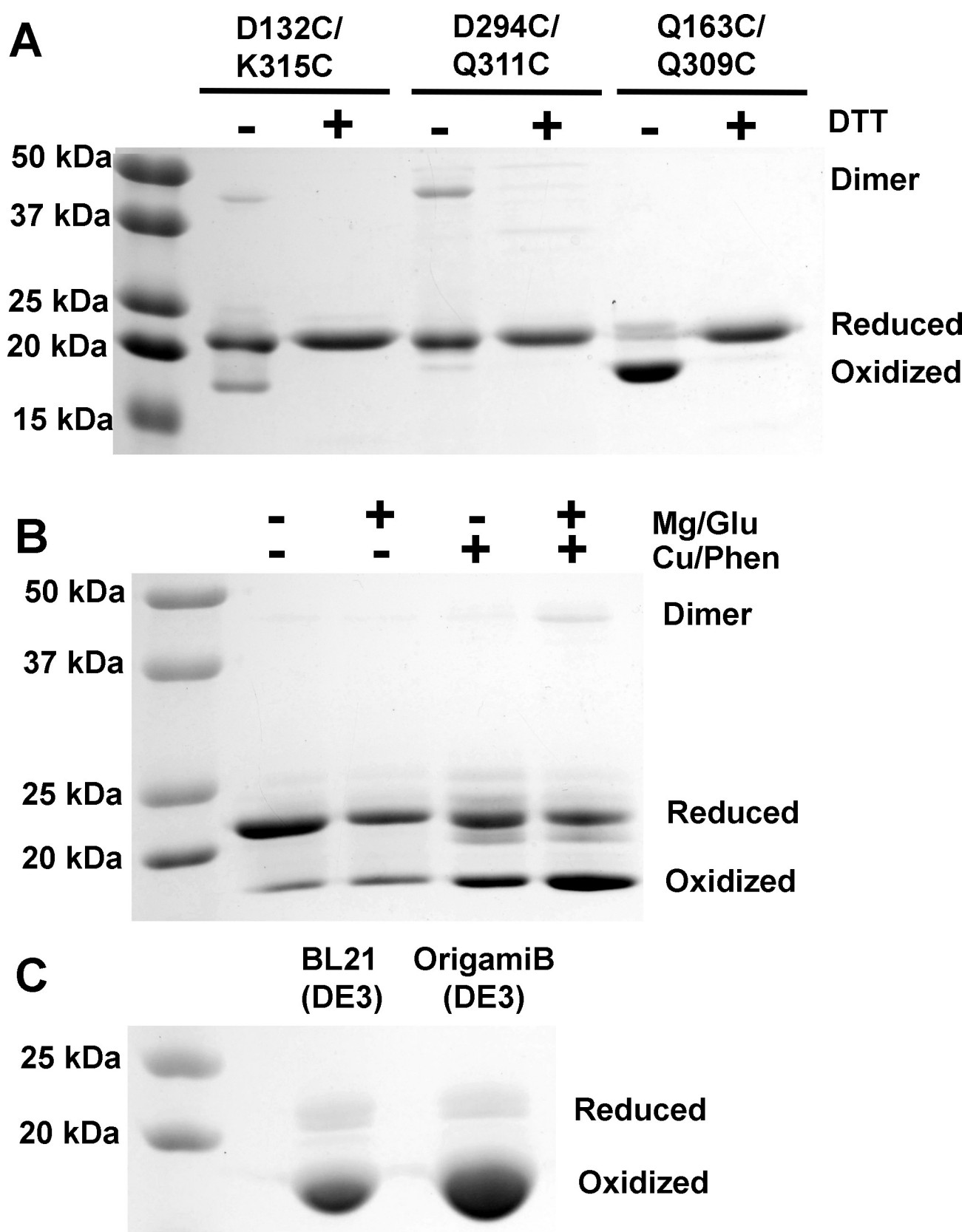

**Fig 1. Expression of internally disulfide bonded mutants of $\alpha_M$I-domain.** A) The SDS-PAGE analysis of three disulfide bond mutants produced in BL21(DE3) and the effect of DTT on their migration. B) The SDS-PAGE analysis of the effect of 0.1 μM Cu²⁺/phenanthroline complex, 1 mM Mg²⁺, and 10 mM glutamate on the formation of the intramolecular disulfide bond in the D132C/K315C mutant. C) SDS-PAGE analysis of the Q163C/Q309C mutant produced in BL21(DE3) and OrigamiB(DE3).

stabilize the active conformation. Fig 1B shows the result of the experiment. Not surprisingly, 1 mM MgCl₂ and 10 mM sodium glutamate had little effect on the formation of the intramolecular disulfide bond by themselves, but when used in combination with 0.1 μM Cu²⁺/phenanthroline, they increased the fraction of the proteins with the intramolecular disulfide bond to ~ 50%. However, many other minor bands as well as a band corresponding to the dimer were also visible. We think the minor bands may be produced by the oxidation of other amino acids by Cu²⁺. Interestingly, the original study of the D132C/K315C mutant noted that the protein could not be successfully crystallized [11]. This may be a consequence of the heterogeneous oxidation of the protein.

Because the intramolecular environment of *E. coli* is reducing and not conducive to the formation of disulfide bonds, we also examined whether the engineered *E. coli* strain OrigamiB (DE3), which contains mutations in its genes for glutathione reductase and thioredoxin reductase, can produce more proteins with the intramolecular disulfide bond. Fig 1C shows the SDS-PAGE of the Q163C/Q309C mutant produced in BL21(DE3) and OrigamiB(DE3). Densitometry analysis of band intensities showed the OrigamiB(DE3) strain modestly improved the amount of protein with the intramolecular disulfide bond from ~ 90% to ~ 95%.

## Yield and thermal stability of active mutants

Compared with the I316G and ΔK315 mutants, the yield of Q163C/Q309C mutant is significantly higher. Fig 2A shows the final Superdex 75 chromatograms of Q163C/Q309C, I316G, and ΔK315 mutants. Based on the area under the elution peak, we estimate the yield of the Q163C/Q309C mutant was ~ 3.5 times that of the I316G and ΔK315 mutants (382 mAU*mL/L for the Q163C/Q309C mutant vs 90 mAU*mL/L for ΔK315 and 87 mAU*mL/L for I316G). BCA assay of the final products showed that the yield of the Q163C/Q309C mutant was ~ 3.8 mg / L whereas the yield for the I316G and ΔK315 mutants was ~ 1.1 mg / L. To compare the stability of active $\alpha_M$I-domain mutants, we carried out differential scanning fluorimetry [20]. The results show that the Q163C/Q309C mutant is significantly more stable than the I316G and ΔK315 mutants. In particular, while the Q163C/Q309C mutant has a melting temperature of ~ 61°C, the melting temperatures of the I316G and ΔK315 mutants were only 48°C and 45°C, respectively. Interestingly, the melting temperature of the Q163C/Q309C mutant is slightly higher than the wild type $\alpha_M$I-domain (58°C). This indicates the disulfide bond adds significant stability to the protein structure.

## SPR analysis of active $\alpha_M$I-domain's interaction with C3d

Although HEK293T cells expressing Mac-1 with the Q163C/Q309C mutations were shown to have a higher affinity for ligands than wild type Mac-1 expressing cells [12], there is no biochemical study showing the Q163C/Q309C $\alpha_M$I-domain has enhanced affinity for ligands compared to wild type I-domain. To confirm that the Q163C/Q309C mutant indeed has enhanced affinity for ligands, we examined the mutant's interactions with C3d, the Mac-1-binding domain in complement 3 [6]. Fig 3 shows the SPR sensorgrams for the $\alpha_M$I-domain variants flowed over a sensor with immobilized C3d. Fitting the equilibrium portions of these sensorgrams to a one-to-one binding model showed that, Q163C/Q309C, ΔK315, and I316G mutants all have dissociation constant of binding (Kd) in the 0.5 to 1 μM range, similar to

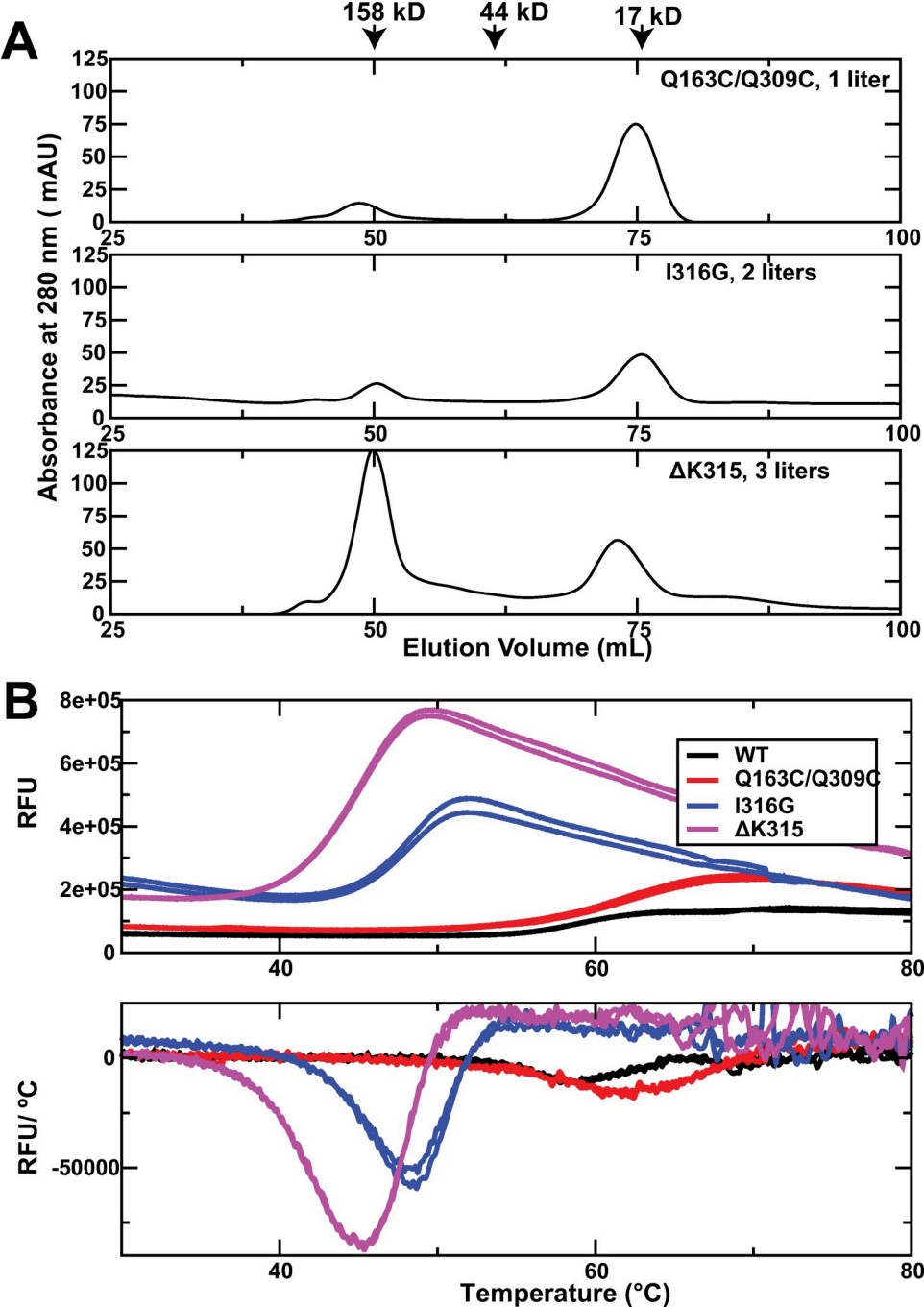

**Fig 2. Comparison of the yield and stability of the active mutants.** A) The final SEC chromatograms of the three active mutants. The chromatograms are proteins from 1 liter of the Q163C/Q309C mutant, 2 liters of the I316G mutant, and 3 liters of the ΔK315 mutant. B) Fluorescence changes in the thermal shift assay of the active mutants. The top panel is the plot of the fluorescence signal against the temperature. The bottom panel is the plot of the derivative of the fluorescence signal with respect to the temperature.

those reported in other studies [6, 7]. These results indicate the Q163C/Q309C mutant has a similar affinity for C3d as other active mutants. However, it is notable that the $k_{off}$ rate of ΔK315 appears to be slower than the other two active mutants, implying the kinetics of binding

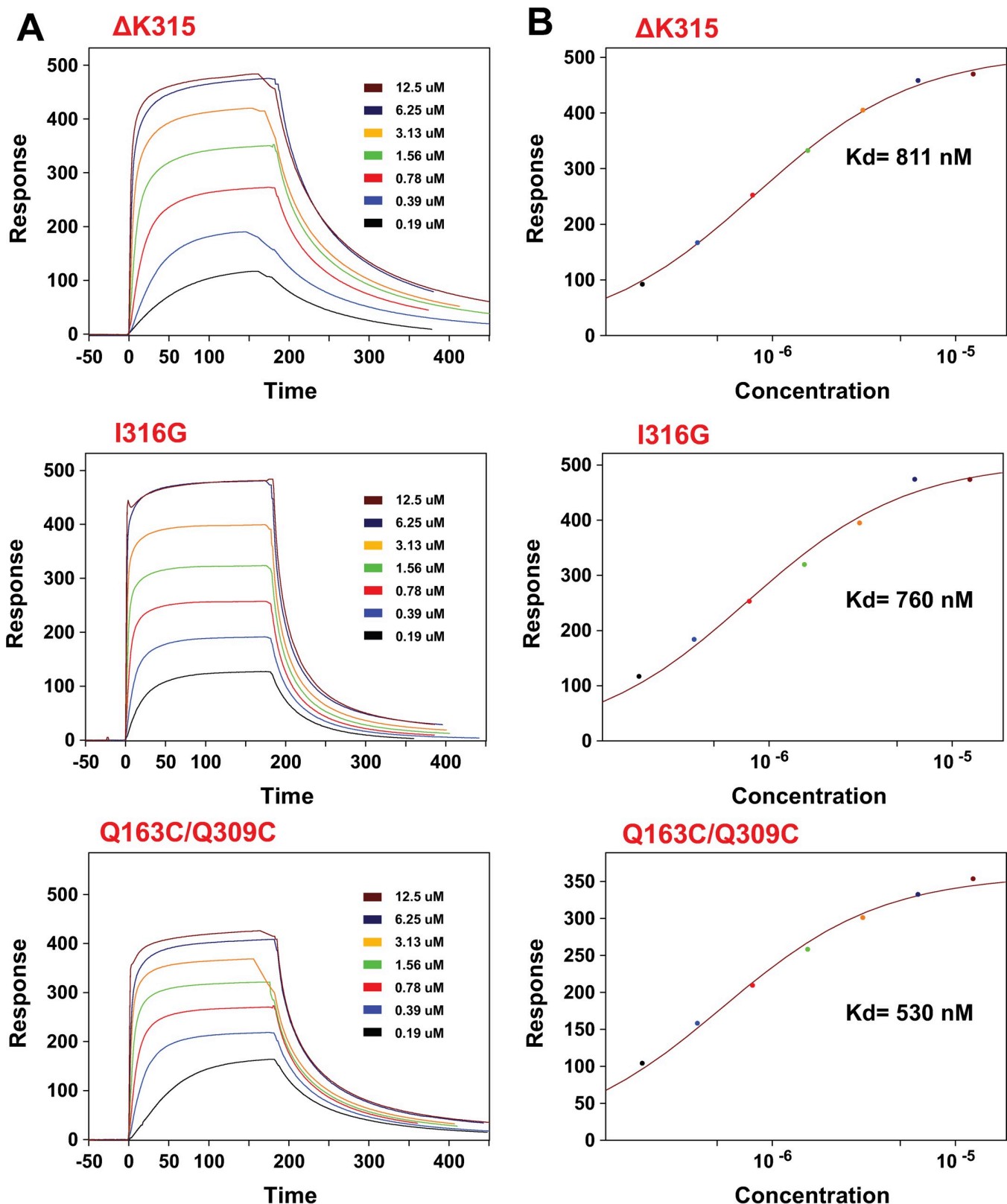

**Fig 3. SPR characterization of C3d's interactions with active α$_M$I-domains.** A) sensorgrams of a C3d-functionalized sensor treated with 0.19, 0.39 0.78, 1.56, 3.13, 6.25, and 12.5 µM of ΔK315, I316G, or Q163C/Q309C mutants. Injection of α$_M$I-domain started at time zero and ended after 180 seconds. B) Estimation

of the dissociation constant (Kd) of the interaction for each active $\alpha_M$I-domain by fitting the equilibrium values of the response curves to a one-to-one binding model.

differ slightly among the mutants. Control experiments carried out using inactive $\alpha_M$I-domain showed inactive $\alpha_M$I-domain does not have a measurable affinity for C3d (Fig 2 of S1 Fig).

### Solution NMR analysis of the Q163C/Q309C mutant

To investigate whether the Q163C/Q309C mutant has better NMR spectral quality than other active mutants, we collected the $^{15}$N-edited HSQC spectra of 0.1 mM $^{15}$N-labeled ΔK315, I316G, and Q163C/Q309C mutants (Fig 4A). A comparison of the spectra showed the Q163C/Q309C mutant produced significantly higher signal intensities than the other two mutants. In addition, samples of the ΔK315 and I316G mutants showed large signal intensity decreases after 24 hours at room temperature whereas the Q163C/Q309C sample of the same

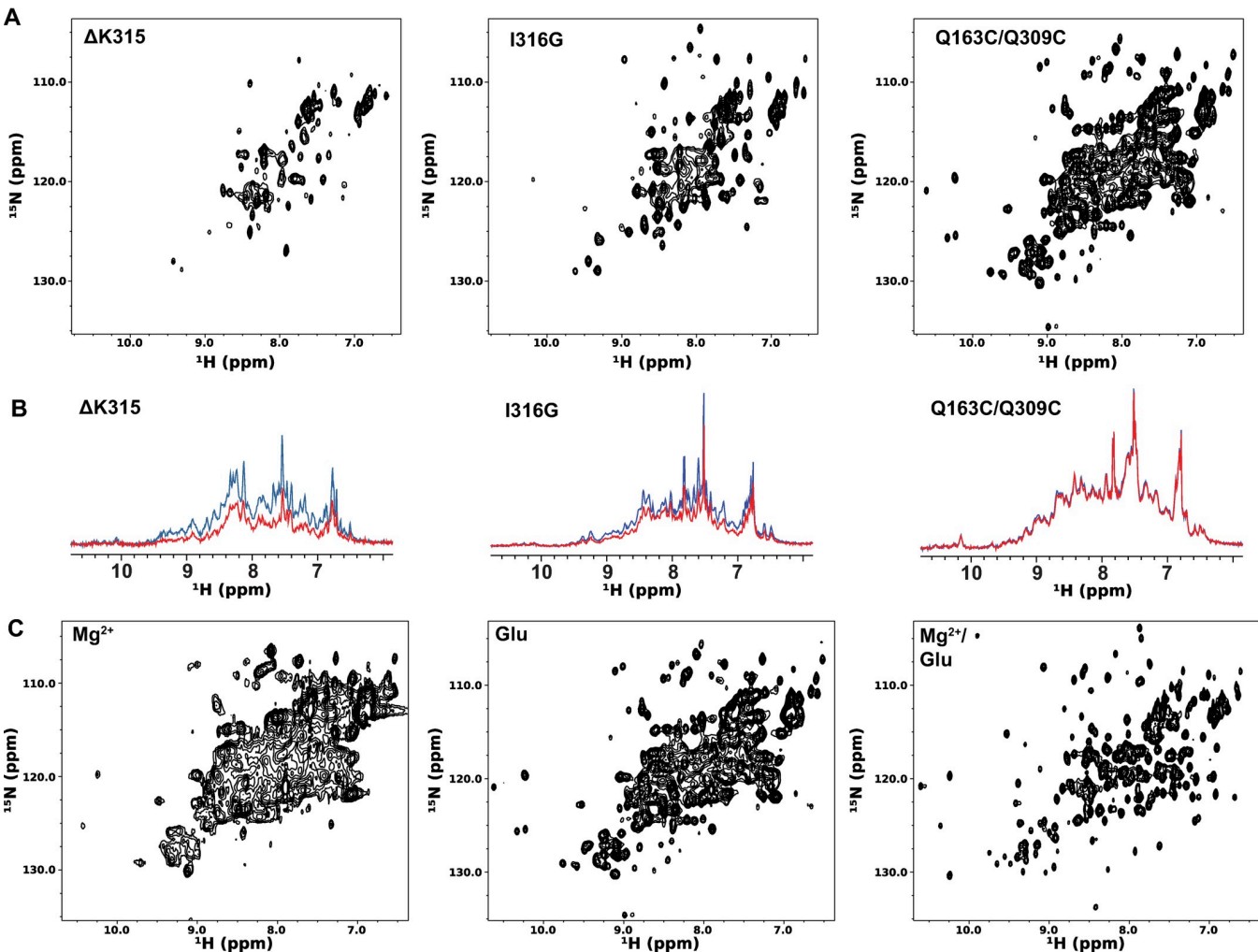

**Fig 4. $^{15}$N-edited HSQC spectra of active mutants.** A) Comparison of the $^{15}$N-edited HSQC spectra of 0.1 mM ΔK315, I316G, and Q163C/Q309C mutants. Spectra were acquired with identical parameters and displayed at the same contour level. B) 1$^{st}$ increment of the $^{15}$N-edited HSQC spectra of 0.1 mM ΔK315, I316G, and Q163C/Q309C mutants before (blue) and after (red) a 24-hour incubation at room temperature. C) Effects of 1 mM MgCl$_2$ and 10 mM sodium glutamate on the spectrum of the Q163C/Q309C mutant.

concentration showed no significant change (Fig 4B). These results indicate the Q163C/Q309C mutant is more soluble and stable than the ΔK315 and I316G mutants.

To confirm that the Q163C/Q309C mutant can bind ligands using the metal-mediated mechanism, we also investigated how $Mg^{2+}$ and glutamate affect the NMR spectrum of the mutant. As shown in Fig 4C, glutamate produced no changes in the HSQC spectrum of the mutant while the addition of $MgCl_2$ reduced the signal intensity without producing new signals. We attribute this to the fact that, in the presence of $MgCl_2$, the active $α_M$I-domain has a significantly higher affinity for carboxyl-containing molecules, and this may have led to the formation of large homo-oligomers that are not detectable by solution NMR. Consistent with this hypothesis is the fact that the presence of both glutamate and $MgCl_2$ resulted in large chemical shift changes and significantly stronger signals. We think this is because glutamate, by acting as the competing ligand, was able to dissociate the homo-oligomers, thereby producing the spectrum of the Q163C/Q309C mutant in the ligand-chelated form.

To verify that the conformation of the Q163C/Q309C mutant is similar to the active $α_M$I-domain and not the inactive $α_M$I-domain, we measured pseudocontact shifts (PCS) induced by the paramagnetic ion $Co^{2+}$. $α_M$I-domain naturally binds $Co^{2+}$, and $Co^{2+}$ does not affect the activity of the protein [21]. This allows the PCS induced by $Co^{2+}$ to be used to validate the structure of the Q163C/Q309C mutant. PCS arise from the dipole-dipole interactions between a paramagnetic metal ion with an anisotropic magnetic susceptibility tensor and nearby atoms. It is both distance and orientation dependent. As a result, PCS has become a valuable tool in protein structure validation [22]. We assigned some backbone amide hydrogen and nitrogen chemical shifts for both the $Co^{2+}$ and $Mg^{2+}$ species of the Q163C/Q309C mutant in the presence of 10 mM glutamate (see Fig 3 of S1 Fig for sample spectral data). In total, 77 out of 186 non-proline residues in the $Mg^{2+}$ species were assigned. Out of the 77 assigned residues, assignments for 56 of which were made for the $Co^{2+}$ species. This allowed us to extract 38 backbone amide hydrogen PCS with magnitudes larger than 0.05 ppm (S1 Table). Fitting the PCS values to the active $α_M$I-domain structure (PDB 1IDO) produced a best-fitting magnetic susceptibility tensor with a paramagnetic center less than 0.9 Å away from the position of the metal in the crystal structure (Fig 5). The agreement between experimental and predicted PCS values is also excellent, with a quality factor of 0.04 (Fig 5). However, the same set of PCS did not fit as well to the inactive structure of $α_M$I-domain (PDB 1JLM). In particular, the paramagnetic center of the best-fitting magnetic susceptibility tensor for the inactive structure is more than 7 Å away from the metal in the crystal structure (Fig 5) and the quality factor of fitting was 0.16, significantly larger than that of the fitting to the active structure. These results support the conclusion that the Q163C/Q309C mutant adopts the active conformation.

## Carboxymethyl sepharose purification of the Q163C/Q309C mutant

To provide proteins of the highest purity for structural biology studies, we also devised a strategy to remove the minor population of reduced protein. The method takes advantage of the fact that $α_M$I-domain in the active conformation has a significantly higher affinity for carboxyl-containing amino acids such as glutamate than in the inactive conformation [19]. Based on this, we postulated that carboxymethyl (CM) sepharose chromatography resin may have a higher affinity for the active $α_M$I-domain than the inactive $α_M$I-domain in the presence of $MgCl_2$, thereby separating the active $α_M$I-domain from the inactive $α_M$I-domain. Fig 6A shows the CM column chromatogram of the Q163C/Q309C mutant. The protein was loaded onto the column in 20 mM HEPES, 0.15 M NaCl, and 2 mM $MgCl_2$, pH 7.0. The column was washed with five column volumes of the same buffer after sample application. Finally, the mobile phase was switched to 20 mM HEPES, 0.15 M NaCl, and 10 mM EDTA, pH 7.0. The

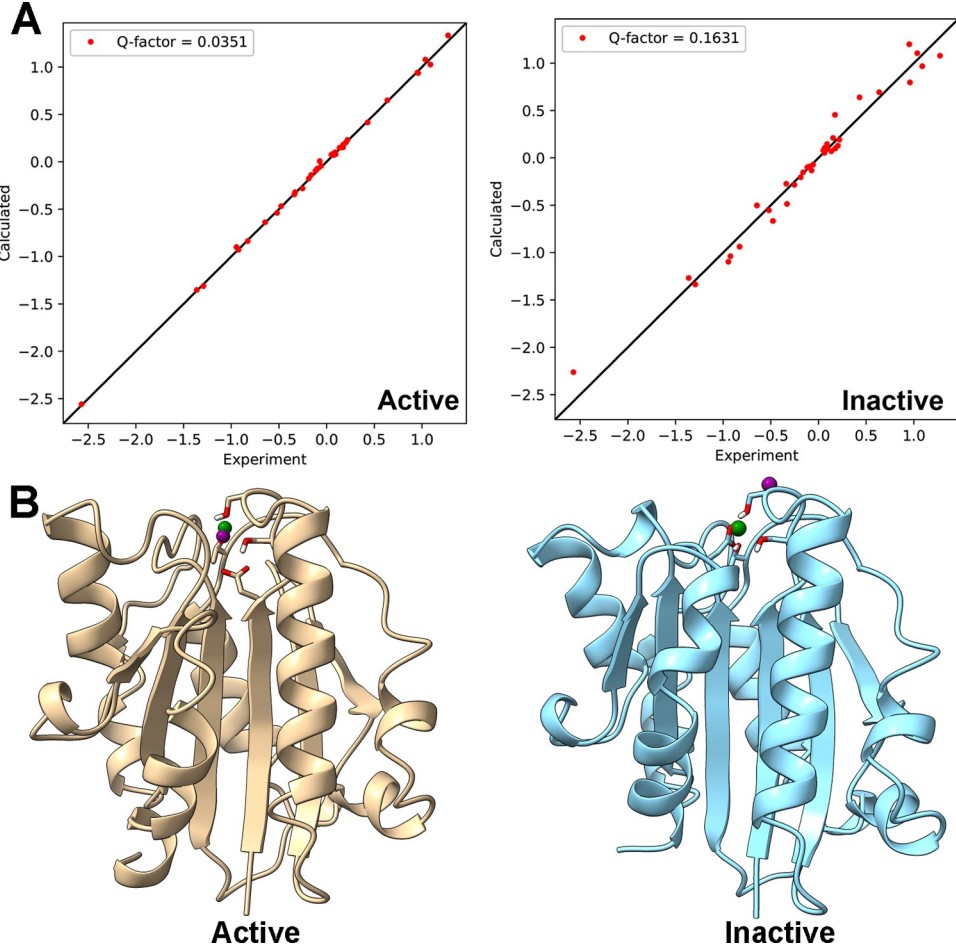

**Fig 5. Consistency of the Q163C/Q309C mutant's Co$^{2+}$-induced PCS with crystal structures of active and inactive α$_M$I-domains.** A) Plots of calculated vs experimental PCS. Calculated PCS were predicted from the best-fit magnetic susceptibility tensor obtained using the structure of either active α$_M$I-domain (PDB: 1IDO) or inactive α$_M$I-domain (PDB: 1JLM). B) Ribbon representations of active (tan) and inactive (cyan) α$_M$I-domain with the position of the metal in the crystal structure indicated by a green sphere. The PCS-predicted paramagnetic center indicated by a purple sphere.

chromatogram showed that a small peak emerged during sample application while two larger peaks were seen after the application of the EDTA buffer. SDS-PAGE analysis showed that the flow through (FT) peak contained a ~ 50% / 50% mixture of reduced and disulfide bonded species (Fig 6A) whereas the elution peaks contained only the disulfide bonded form of the protein. This result is consistent with our postulate that the active form of the protein has a higher affinity for CM resin in the presence of MgCl$_2$.

We also studied the proteins in the two elution peaks using solution NMR. Surprisingly, the protein in the first elution peak resembled that of the protein in the presence of both MgCl$_2$ and glutamate, signifying that this fraction of the protein may be chelating a carboxyl-containing ligand (Fig 6B). Protein in the second elution peak produced a spectrum identical to that of the apo Q163C/Q309C mutant (Fig 6C). The addition of EDTA changed the spectrum of the protein in elution peak 1 to that of the apo Q163C/Q309C mutant (Fig 6D), confirming that elution peak 1 contained active α$_M$I-domain bound to ligands through divalent cation mediated interactions.

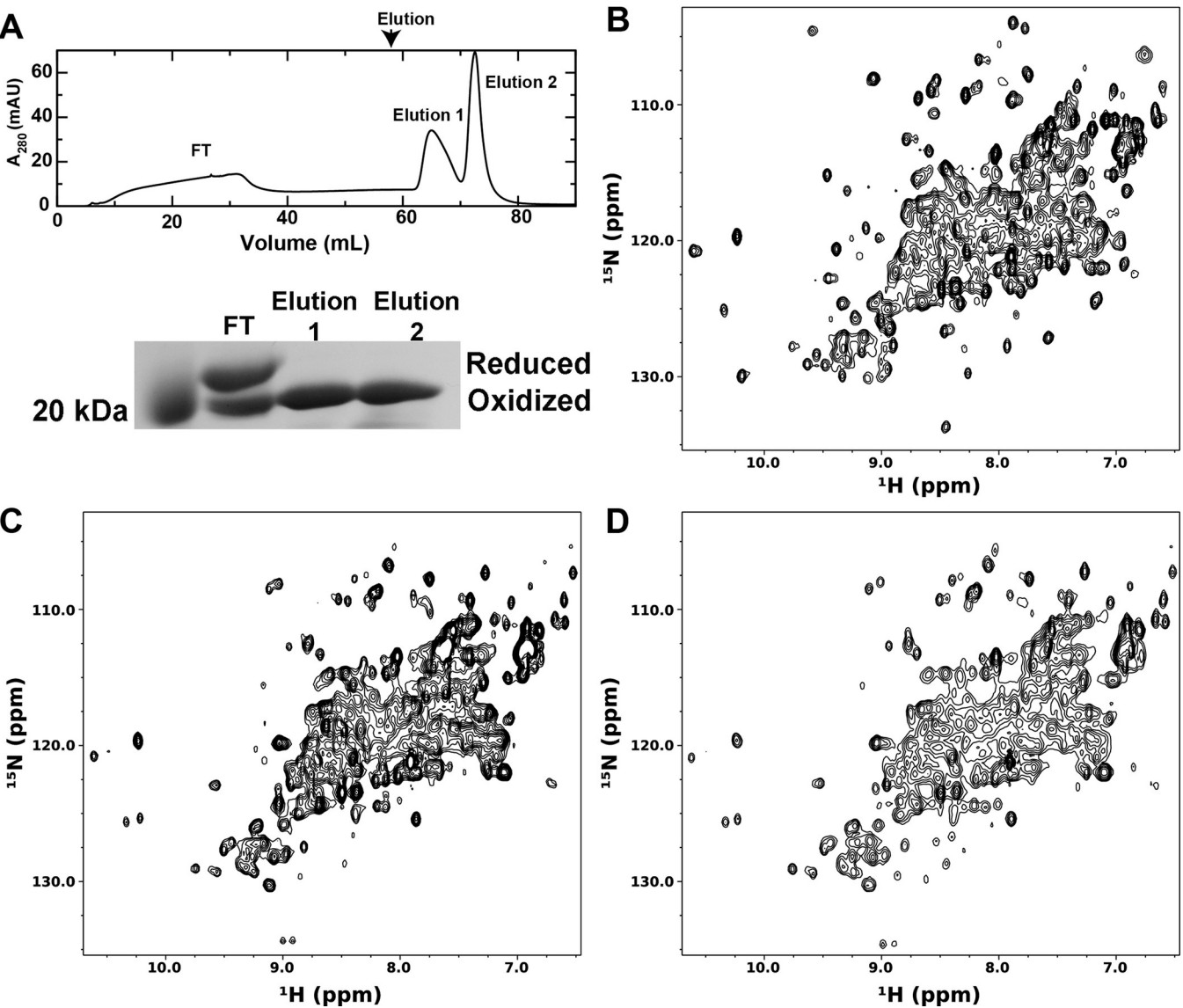

**Fig 6. CM purification of the Q163C/Q309C mutant.** A) The CM-sepharose chromatogram produced by the Q163C/Q309C mutant and the SDS-PAGE analysis of the flow through and elution fractions. Note that no reducing agent was used in the SDS-PAGE. B) $^{15}$N-edited HSQC spectrum of the protein in elution peak 1 before EDTA treatment. C) $^{15}$N-edited HSQC spectrum of the protein in elution peak 1 after EDTA treatment. D) $^{15}$N-edited HSQC spectrum of the protein in elution peak 2.

## Discussion

Understanding the interactions between integrins and their ligands is central to the understanding of integrin activities. This is especially true for integrin Mac-1, whose ligands possess diverse structures and physical properties. Although $\alpha_M$I-domain's interactions with ligands have been studied using X-ray crystallography, the lack of an active $\alpha_M$I-domain suitable for solution NMR has prevented such interactions from being characterized by solution NMR, a powerful and versatile technique that has much to offer in way of characterizing protein-ligand interactions. In this report, we examined five active mutants for their suitability in NMR studies. We were especially interested in mutants that used an intramolecular disulfide bond to constrain the position of the C-terminal α7 helix rather than mutants that disrupted the

hydrophobic core of $\alpha_M$I-domain. We thought such an activation mechanism should produce more stable and soluble proteins than strategies that disrupt the hydrophobic core of the protein.

Out of the three disulfide bonded mutants, only the Q163C/Q309C mutant formed the intramolecular disulfide bond spontaneously. Characterization of the protein's yield, stability, and NMR spectral quality showed the protein was considerably more stable than the ΔK315 and I316G mutants, the two most commonly used active variants of $\alpha_M$I-domain. It also produced superior NMR spectra and has a higher yield than the other two mutants. In addition, $Co^{2+}$-induced PCS data showed the conformation of the Q163C/Q309C mutant is consistent with active $\alpha_M$I-domain but not inactive $\alpha_M$I-domain. This result is in agreement with SPR data that indicate the Q163C/Q309C mutant has a similar affinity for C3d as the ΔK315 and I316G mutants.

Although treatment with oxidizing agents such as $Cu^{2+}$/phenanthroline has been shown to induce the formation of an intramolecular disulfide bond in the D132C/K315C mutant, in our hands the treatment also produced several other species that could have resulted from the non-specific oxidation of amino acids such as tyrosine and arginine, a possible factor in the unsuccessful crystallization of the protein after oxidation treatment [11]. It was interesting that the addition of $MgCl_2$ and glutamate, compounds that bind $\alpha_M$I-domain through the MIDAS of active $\alpha_M$I-domain, was able to significantly improve the amount of disulfide bonded D132C/K315C mutant. This shows that stabilizing the protein in the active conformation may help the formation of intramolecular disulfide bonds.

To improve the yield of the Q163C/Q309C mutant, we employed two approaches. First, we examined whether expressing the protein in OrigamiB(DE3), which contains lower levels of the reducing agent glutathione and thioredoxin, increases the fraction of disulfide bonded protein. The result showed OrigamiB(DE3) increased the amount of disulfide bonded protein modestly from ~ 90% to ~ 95%. Second, to remove the remaining protein with no disulfide bond, we leveraged active $\alpha_M$I-domain's affinity for carboxyl-containing compounds to separate the active fraction from the inactive fraction using CM chromatography. This works well as the reduced protein was not retained in the column. However, a fraction of the active protein was also found in the flow through. One explanation is that, in the presence of $MgCl_2$, inactive $\alpha_M$I-domain may be bound to active $\alpha_M$I-domain because of the latter's high affinity for glutamate, resulting in a complex incapable of binding the resin. The formation of this inactive-active $\alpha_M$I-domain heterodimer may explain why an equal amount of active $\alpha_M$I-domain was also found in the flow-through fractions. This postulate is corroborated by the observation that a glutamate from one $\alpha_M$I-domain chelated the divalent cation in the MIDAS of another $\alpha_M$I-domain in the crystal structure of the ΔK315 mutant [8]. We think that it may be possible to minimize the amount of disulfide bonded proteins in the flow through if the protein is diluted to a low concentration to prevent the formation of homo-oligomers before applying to the column. Another unexpected outcome is that some of the proteins in the elution are consistent with ligand-bound forms of $\alpha_M$I-domain. It is unclear what the ligand is. One possibility is that the ligand may be contaminating small CM oligosaccharides that were inadvertently extracted by the protein.

## Conclusion

The integrin Mac-1 is an important integrin involved in many aspects of leukocyte biology. Understanding the mechanisms of its ligand specificity is essential to developing targeted treatments against Mac-1. However, the lack of a solution NMR suitable active $\alpha_M$I-domain has so far prevented NMR from investigating the interactions of active $\alpha_M$I-domain with its

ligands. This report systematically examined five known active mutants of $\alpha_M$I-domain and showed that the Q163C/Q309C mutant adopts the active conformation, can be produced with higher yield, and is more stable than other commonly used active mutants of $\alpha_M$I-domain. The availability of such a mutant will enable more NMR studies of $\alpha_M$I-domain's interaction with ligands and reveal more insights into the mechanisms of Mac-1 activity.

## Supporting information

**S1 Fig. Supporting figures and unmodified gel pictures.**
(DOCX)

**S1 Table. Table of backbone amide hydrogen PCS greater than 0.05 ppm.**
(DOCX)

## Acknowledgments

We thank Drs. Brian Cherry and Samrat Amin in ASU's Magnetic Resonance Research Center for maintenance of the NMR instruments. We also want to thank Ms. Shundene Key for performing the thermal shift assays.

## Author Contributions

**Conceptualization:** Hoa Nguyen, Xu Wang.

**Data curation:** Hoa Nguyen, Xu Wang.

**Formal analysis:** Hoa Nguyen, Tianwei Jing, Xu Wang.

**Funding acquisition:** Xu Wang.

**Investigation:** Hoa Nguyen, Xu Wang.

**Methodology:** Tianwei Jing, Xu Wang.

**Project administration:** Xu Wang.

**Resources:** Tianwei Jing, Xu Wang.

**Software:** Tianwei Jing, Xu Wang.

**Supervision:** Xu Wang.

**Validation:** Xu Wang.

**Visualization:** Xu Wang.

**Writing – original draft:** Xu Wang.

**Writing – review & editing:** Hoa Nguyen, Xu Wang.

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
