## [Decision Letter · Decision Letter 0]

24 Oct 2022

PONE-D-22-26948The Q163C/Q309C Mutant of αMI-domain is an Active Variant Suitable for NMR CharacterizationPLOS ONE

Dear Dr. X. Wang,

Thank you for submitting your manuscript to PLOS ONE. As you can see below both Reviewers agree on the good impact of your manuscript that is well and clearly presented.However, they highlight some minor points that require improvement prior to acceptance.Please note that, in particular, it would be very appreciated if your experimental data will be deposited and a full account of data analysis process with representative HNCACB and HNCOCACB spectra will be included as supplementary material.

We look forward to receiving your revised manuscript.

Kind Regards

Matteo De March

Academic Editor

PLOS ONE

“This study was supported by funding from NIH/NIGMS (R01GM118518). We thank Drs. Brian Cherry and Samrat Amin in ASU’s Magnetic Resonance Research Center for maintenance of the NMR instruments.”

“The study was funded by a grant from the National Institutes of Health to XW (www.nih.gov, grant #R01GM118518). NIH did not play any role in the study design, data collection and analysis, decision to publish, or preparation of the manuscript.”

“NO authors have competing interests”

Reviewers' comments:

Reviewer's Responses to Questions

**Comments to the Author**

1. Is the manuscript technically sound, and do the data support the conclusions?

Reviewer #1: Yes

Reviewer #2: Yes

2. Has the statistical analysis been performed appropriately and rigorously? 

Reviewer #1: N/A

Reviewer #2: N/A

3. Have the authors made all data underlying the findings in their manuscript fully available?

Reviewer #1: No

Reviewer #2: No

4. Is the manuscript presented in an intelligible fashion and written in standard English?

Reviewer #1: Yes

Reviewer #2: Yes

5. Review Comments to the Author

Reviewer #1: This manuscript is an elegant example of rational engineering to retain function, and enable structural/physical analysis. The goal is to study an ligand-binding domain of an integrin, in the active state. Previous work proposes mutations to "stabilize" the domain, but here it is shown that such mutations do not stabilize the domain. The authors propose new mutations that would lead to an active state whereby they would stabilize with an intramolecular disulphide. All the data presented supports the conclusion of a stable active domain, although it requires glutamate/Mg2+ to maintain a state suitable for NMR analysis. The paper is well written, figures are clear. The only issue is a description of the Co+-induced pseudocontact shifts are based on the assignments. I would like these assignments to be deposited in the BMRB or if you have state the accession number. I consider this important as the BMRB will assess the quality (accuracy) of assignment. I do agree the fitting of the PCS is convincing that all is good, but you need to deposit the data.

Reviewer #2: This very clearly written and presented MS makes a strong and logically argued case for the use of an engineered domain, from a key integrin (CR3), for ligand-binding and screening studies, and in particular for studies using NMR spectroscopy. The modified domain has two non-native cysteines that form a disulfide bond, presumably stabilising an active conformation of the domain that is able to bind ligands (unlike the wild type domain) via a MIDAS motif. Because of the many reported ligands for this domain and its well-established central role in immune responses, the topic is of quite wide interest.

It would be good to show (eg in supplementary) the position of the mutated residues and perhaps a predicted structure of the disulfide-containing molecule.

Readers might be a little sceptical of how the authors were able to assign the amide nitrogens and protons for this mutant given the sub-optimal nature of the HSQC spectrum that contains many, well dispersed but very small peaks but also a region of more intense and overlapping peaks in the random-coil region. To reassure us, they should include in supplementary data a full account of this process including a method section and representative HNCACB and HNCOCACB spectra, and information regarding the extent of 15NH assignments (that should be deposited in a database) for both Mg and Co forms.

Another improvement to this MS would be to include some KD values based on SPR studies using a dilution series of their domains (binding to immobilised C3d). The authors seem to have plenty of sample for the binding partners and access to the kit so it is puzzling why they have not already included such measurements. It looks like the alpha-helix truncation mutant might bind to C3d more tightly than the double-Q (disulfide) mutant since it has a slower off-rate. That does not really weaken their conclusions but is good to know.

Small things:

Understanding.... is... (not are)

Data are... (not is)

And they seem to use the terms intracellular and intramolecular indiscriminately - I think they mean intermolecular (?)

6. PLOS authors have the option to publish the peer review history of their article (what does this mean?). If published, this will include your full peer review and any attached files.

Reviewer #1: No

Reviewer #2: **Yes: **Paul N Barlow

---

## [Author Response · Author response to Decision Letter 0]

8 Dec 2022

Reviewer #1: This manuscript is an elegant example of rational engineering to retain function, and enable structural/physical analysis. The goal is to study an ligand-binding domain of an integ-rin, in the active state. Previous work proposes mutations to "stabilize" the domain, but here it is shown that such mutations do not stabilize the domain. The authors propose new mutations that would lead to an active state whereby they would stabilize with an intramolecular disulphide. All the data presented supports the conclusion of a stable active domain, although it requires gluta-mate/Mg2+ to maintain a state suitable for NMR analysis. The paper is well written, figures are clear. The only issue is a description of the Co+-induced pseudocontact shifts are based on the assignments. I would like these assignments to be deposited in the BMRB or if you have state the accession number. I consider this important as the BMRB will assess the quality (accuracy) of assignment. I do agree the fitting of the PCS is convincing that all is good, but you need to de-posit the data.

Reply: The chemical shifts of both the Mg2+ and Co2+ species of the Q163C/Q309C bound to glutamate have been deposited with BMRB. The accession numbers are 51714 for the Mg2+ spe-cies and 51716 for the Co2+ species. A detailed description of the assignment process is also in-cluded in the manuscript (lines 162 to 173). In particular, the Mg2+ species were assigned with HNCACB and HNCOCACB using conventional methods. To assign the Co2+ species, we used HNCACB and HNCOCACB of the Co2+ sample and the fact that PCS values are similar for am-ide hydrogen and nitrogen pairs if the atoms are not very close to the paramagnetic center to pre-dict possible assignments. The proposed assignments were verified using CA and CB chemical shifts.

Reviewer #2: This very clearly written and presented MS makes a strong and logically argued case for the use of an engineered domain, from a key integrin (CR3), for ligand-binding and screening studies, and in particular for studies using NMR spectroscopy. The modified domain has two non-native cysteines that form a disulfide bond, presumably stabilising an active con-formation of the domain that is able to bind ligands (unlike the wild type domain) via a MIDAS motif. Because of the many reported ligands for this domain and its well-established central role in immune responses, the topic is of quite wide interest.

It would be good to show (eg in supplementary) the position of the mutated residues and perhaps a predicted structure of the disulfide-containing molecule.

Reply: Models of the disulfide bond mutants have been added to the supporting information (Figure 1 in S1 figures).

Readers might be a little sceptical of how the authors were able to assign the amide nitrogens and protons for this mutant given the sub-optimal nature of the HSQC spectrum that contains many, well dispersed but very small peaks but also a region of more intense and overlapping peaks in the random-coil region. To reassure us, they should include in supplementary data a full account of this process including a method section and representative HNCACB and HNCOCACB spec-tra, and information regarding the extent of 15NH assignments (that should be deposited in a database) for both Mg and Co forms.

Reply: We apologize for neglecting to mention the details of our assignment process. In particu-lar, we hope we have not given the false impression that all residues in the two metal-bound forms were assigned. Because our goal is to validate the structure, we only obtained enough as-signments to allow the evaluation of the structure of the Q163C/Q309C mutant. As for the quali-ty of the spectra, 3D spectra were collected using samples containing 1 mM divalent cation and 10 mM glutamate. These ligands significantly improved the spectral quality of the Q163C/Q309C mutant. This was mentioned in the methods section, and now we also added the detail to the results section. The samples were also deuterated to reduce relaxation and increase intensity. We have included projections of the HNCACB and HNCOCACB data in the support-ing information (Figure 3 in S1 figure). Assignments have also been deposited with BMRB (ac-cession code 51714 for the Mg2+ species and 51716 for the Co2+ species). A detailed description of the assignment process has been included in the methods section (lines 162 to 173). In total, 77 residues out of 186 non-proline residues in the Mg2+ form were assigned. Out of these 77 res-idues, 56 residues in the Co2+ form were assigned, 38 of which had HN PCS greater than 0.05 ppm.

Another improvement to this MS would be to include some KD values based on SPR studies us-ing a dilution series of their domains (binding to immobilised C3d). The authors seem to have plenty of sample for the binding partners and access to the kit so it is puzzling why they have not already included such measurements. It looks like the alpha-helix truncation mutant might bind to C3d more tightly than the double-Q (disulfide) mutant since it has a slower off-rate. That does not really weaken their conclusions but is good to know.

Reply: We were unable to include full titration series for these active variants in our initial manu-script because non-specific interactions of active mutants with the sensor prevented the inclusion of data acquired with higher concentrations of the mutants. We have repeated these SPR experi-ments with lower concentrations of mutants and the data are now included in the manuscript. Kds obtained from equilibrium values of the responses at different concentrations showed the affinities of these mutants for C3d are similar and in agreement with reported values (lines 255-269, Figure 3). These data also confirmed what the reviewer already noted, that is the dissocia-tion rate of the ΔK315 mutant is slower than the other mutants. However, we were not able to obtain good fittings of the association and dissociation phases to calculate the actual rates.

Small things:

Understanding.... is... (not are)

Data are... (not is)

And they seem to use the terms intracellular and intramolecular indiscriminately - I think they mean intermolecular (?)

Reply: We have carefully gone through the manuscript to correct all grammatical mistakes we can find. All mentions of “intracellular” have been replaced with “intramolecular”.

---

## [Decision Letter · Decision Letter 1]

4 Jan 2023

PONE-D-22-26948R1

The Q163C/Q309C Mutant of αMI-domain is an Active Variant Suitable for NMR Characterization

PLOS ONE

Dear Dr. Wang,

Thank you for submitting your manuscript to PLOS ONE. After careful consideration, we feel that it has merit but does not fully meet PLOS ONE’s publication criteria as it currently stands. Therefore, we invite you to submit a revised version of the manuscript that addresses the points raised during the review process.

We look forward to receiving your final version soon.

With Best Regards,

Matteo De March

Academic Editor

PLOS ONE

Journal Requirements:

Additional Editor comments:

your manuscript can be now accepted as both Reviewers agreed with the revised version. However, Reviewer 2 stated "1 AU = 1 pg per mm2 (square mm)". Although it is a very small change, it cannot be ignored.

Reviewers' comments:

Reviewer's Responses to Questions

**Comments to the Author**

1. If the authors have adequately addressed your comments raised in a previous round of review and you feel that this manuscript is now acceptable for publication, you may indicate that here to bypass the “Comments to the Author” section, enter your conflict of interest statement in the “Confidential to Editor” section, and submit your "Accept" recommendation.

Reviewer #1: All comments have been addressed

Reviewer #2: All comments have been addressed

2. Is the manuscript technically sound, and do the data support the conclusions?

Reviewer #1: Yes

Reviewer #2: Yes

3. Has the statistical analysis been performed appropriately and rigorously? 

Reviewer #1: N/A

Reviewer #2: Yes

4. Have the authors made all data underlying the findings in their manuscript fully available?

Reviewer #1: Yes

Reviewer #2: Yes

5. Is the manuscript presented in an intelligible fashion and written in standard English?

Reviewer #1: Yes

Reviewer #2: Yes

6. Review Comments to the Author

Reviewer #1: The authors have addressed comments and questions of the review. The deposition of data to the BMRB is important.

Reviewer #2: The authors have addressed my concerns but please correct statement regarding SPR. You say 1 AU = 1 pg per mm. This should of course be 1 pg per mm2 (square mm)

7. PLOS authors have the option to publish the peer review history of their article (what does this mean?). If published, this will include your full peer review and any attached files.

Reviewer #1: No

Reviewer #2: **Yes: **Paul Nigel Barlow

---

## [Author Response · Author response to Decision Letter 1]

6 Jan 2023

Reviewer #1: The authors have addressed comments and questions of the review. The deposition of data to the BMRB is important.

Response: Thank you for your review.

Reviewer #2: The authors have addressed my concerns but please correct statement regarding SPR. You say 1 AU = 1 pg per mm. This should of course be 1 pg per mm2 (square mm)

Response: Thank you for noticing the mistake in the unit. We have changed it to “1 pg / mm2” (line 149).

---

## [Editor Report · Decision Letter 2]

10 Jan 2023

The Q163C/Q309C Mutant of αMI-domain is an Active Variant Suitable for NMR Characterization

PONE-D-22-26948R2

Dear Prof. X. Wang,

we are pleased to inform you that your manuscript is now ready for publication. Within one week, you’ll receive an e-mail detailing the outstanding technical requirements that need to be addressed before scheduling your publication.

I would like to wish you all the best for your work.

Kind regards.

Matteo De March

Academic Editor

PLOS ONE

---

## [Editor Report · Acceptance letter]

16 Jan 2023

PONE-D-22-26948R2 

The Q163C/Q309C Mutant of αMI-domain is an Active Variant Suitable for NMR Characterization 

Dear Dr. Wang:

I'm pleased to inform you that your manuscript has been deemed suitable for publication in PLOS ONE. Congratulations! Your manuscript is now with our production department. 

Kind regards, 

on behalf of

Dr. Matteo De March 

Academic Editor

PLOS ONE